# Fabrication, Processing, Properties, and Applications of Closed-Cell Aluminum Foams: A Review

**DOI:** 10.3390/ma17030560

**Published:** 2024-01-24

**Authors:** Wensheng Fu, Yanxiang Li

**Affiliations:** 1School of Materials Science and Engineering, Tsinghua University, Beijing 100084, China; fws20@mails.tsinghua.edu.cn; 2Key Laboratory for Advanced Materials Processing Technology, Ministry of Education, Beijing 100084, China

**Keywords:** porous metal, metal foam, aluminum foam, closed-cell foam

## Abstract

Closed-cell aluminum foams have many excellent properties, such as low density, high specific strength, great energy absorption, good sound absorption, electromagnetic shielding, heat and flame insulation, etc. As a new kind of material, closed-cell aluminum foams have been used in lightweight structures, traffic collision protections, sound absorption walls, building decorations, and many other places. In this paper, the recent progress of closed-cell aluminum foams, on fabrication techniques, including the melt foaming method, gas injection foaming method, and powder metallurgy foaming method, and on processing techniques, including powder metallurgy foaming process, two-step foaming process, cast foaming process, gas injection foaming process, mold pressing process, and integral foaming process, are summarized. Properties and applications of closed-cell aluminum foams are discussed based on the mechanical properties and physical properties separately. Special focuses are made on the newly developed cast-forming process for complex 3D parts and the improvement of mechanical properties by the development of small pore size foam fabrication and modification of cell wall microstructures.

## 1. Introduction

There are a variety of porous structures in nature, such as coral, bone, sponge, wood, etc. Inspired by nature, human beings fabricated a series of porous materials [1,2,3,4,5,6,7]. Porous metal has excellent mechanical and physical properties, such as low density, high specific strength, and good sound absorption [8,9,10,11,12,13,14,15,16,17]. Aluminum foams have become the most popular and widely studied porous metal because of their low cost, unique properties, easiness of fabrication, and rich reserves [18,19].

Aluminum foams can be categorized into closed-cell and open-cell foams according to their pore structure [20], as shown in Figure 1. Their structures, properties, and applications are distinct. Open-cell foams are often used as sound-absorbing materials [21] or liquid filtration materials [22]. Detailed descriptions of open-cell foams can be found in [22]. This paper mainly introduces closed-cell foams. In the following content, all the mentioned aluminum foams refer to closed-cell aluminum foams.

The research on aluminum foams can be traced back to 1925 [3,23]. After nearly a hundred years of development, a series of fabrication methods have been developed and commercial production has been achieved [3,11,24,25]. The pore size of foams is generally 1~25 mm, their porosities can reach 98%, and their density is around 0.05~1.3 g/cm^3^; therefore, they are ultra-light materials [8]. Aluminum foams have high strength and great energy absorption capacity, compressive plateau stress $\sigma_{p}$ is around 2~30 MPa, and energy absorption capacity W is around 1~15 MJ/m^3^ [26,27]. Aluminum foams have good sound absorption performance and electromagnetic shielding effectiveness, their sound absorption coefficient can be greater than 0.6 [28], and for the electromagnetic wave of frequency below 200 MHz, the shielding effectiveness can reach 80 dB [29]. Aluminum foams have high porosity and low thermal conductivity [30]. They are not flammable and can be used for heat and flame insulation [10]. Because of these excellent properties, aluminum foams have been used in many places such as sound absorption barriers, architectural decoration, military protection, aerospace, and aeronautics parts. An excellent combination of mechanical and functional properties provides wide prospects for the application of aluminum foams [3,10,31,32].

**Figure 1 materials-17-00560-f001:**
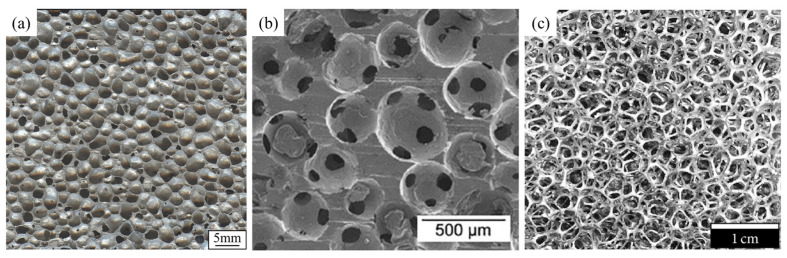
(**a**) Closed-cell aluminum foam [33] (reproduced with permission from Springer); (**b**,**c**) open-cell aluminum foam [34,35] (reproduced with permission from Elsevier and Hindawi).

Fabrication, processing, and properties are the most important research fields of aluminum foams. This paper aims to comprehensively review the fabrication and processing techniques and explore the unique properties and diverse applications of aluminum foams. In addition, the current status of research and development in aluminum foam is discussed. It accomplishes this by providing an overview of existing developments and identifying future opportunities in this industry.

## 2. Fabrication

In decades of research, many fabrication methods were proposed to obtain aluminum foam. The most important and widely used fabrication techniques are the melt foaming (MF) method, gas injection foaming (GIF) method, and powder metallurgy foaming (PMF) method, as these methods have achieved commercial application. The comparison of these three methods is shown in Table 1, the research progress is reviewed in the following content.

### 2.1. Melt Foaming Method

Shinko Wire Company in Japan developed the MF method commercially in 1986 with a trademark of ALPORAS^®^ [3,36]. The ALPORAS process, as shown in Figure 2a, has been widely used since it was invented. The process of this method is to add a thickening agent (TA, usually 1~3 wt% Ca) to the aluminum melt and stir the melt for some time for foam stabilization, then add the foaming agent (FA), quickly stir to disperse it into the melt, and begin to foam [36]. The MF method is suitable for the fabrication of large-size foam blocks and can produce blocks with a length of about 2 m, width of about 1 m, and height of over 0.3 m. The pore size is about 1~8 mm, and the porosity is around 50~90%. After slicing blocks into panels, these panels can be used for sound absorption barriers [37] and architectural decoration [38], as shown in Figure 2b. The disadvantage of this method is its poor shaping ability, as usually only blocks can be fabricated.

To obtain MF foams with uniform pore structures, a lot of studies about the fabrication process have been carried out.

Many studies focused on the thickening and foaming procedure. These studies tried to add different kinds of TAs and FAs to the melt. TAs mainly include Ca [36], Mg [39], fly ash [40], SiC [41], Al powders [42], and MnO_2_ [43]. Among the TAs, Ca shows the best foam stabilization ability, and melt treatment with Ca is the only option for commercial production. Song et al. [44] found that controlling the viscosity of the melt within a certain range by adjusting the content of Ca can obtain good foaming results. FAs mainly include metal hydrides and carbonates [3], for example, TiH_2_ [36], ZrH_2_ [45], CaCO_3_ [41], and MgCa(CO_3_)_2_ [46]. Among these FAs, TiH_2_ is the most widely used and shows the best foaming results. The decomposition temperature of raw TiH_2_ is low (<500 °C) [47,48], far below the foaming temperature (>600 °C), and it is not suitable for foaming. The decomposition temperature of TiH_2_ can be increased by pre-oxidization treatment [49,50] or coating [51] to achieve better foaming results. In these two methods, pre-oxidation treatment is more commonly used. Preheating TiH_2_ at 480~520 °C in an oxidizing atmosphere can increase its decomposition temperature to above 600 °C and fit the foaming temperature [49]. Starting from Ca as the TA and TiH_2_ as the FA, Yuan [52] studied key parameters of the fabrication process, including the stirring process, melt thickening temperature, TiH_2_ particle size, and foaming temperature, and then the realized batch fabrication of 500 × 1000 × *X* mm slabs.

Aluminum foams with smaller pore sizes and uniform pore structure have better mechanical properties (MEPs) [53]. However, it is difficult to fabricate foams with pore diameters ($d_{r}$) smaller than 3 mm by the traditional MF method. The wettability of TiH_2_ in aluminum melt is poor [45], and it is difficult to uniformly disperse TiH_2_ powders into the melt after a few minutes of mechanical stirring before foaming. Although foaming under high pressure [54] or rapidly cooling the sample before the melt is completely foamed [53] can reduce the pore size, these methods are difficult to realize in commercial production. In order to solve this problem, Cheng [33] ball milled the mixed powders of Cu and pre-oxidized TiH_2_. During the mechanical ball milling process, TiH_2_ is broken and embedded into Cu powders, as shown in Figure 3a. Cu powders have better wettability with aluminum melt and are easier to disperse into the melt. Foams with $d_{r}$ of 1.6 mm can be fabricated by using this method. Zhou [55] selected AlMg35 alloy with a low melting point (about 450 °C). Add 5~10 wt% of pre-oxidized TiH_2_ powder into the AlMg35 melt, stir at 500 °C for 15~30 min, and TiH_2_ powders can be uniformly dispersed into the AlMg35 melt. AlMg35 alloy is brittle, and the solidified AlMg35-TiH_2_ composite FA can be easily broken into small pieces, as shown in Figure 3b. Foams with $d_{r}$ of below 1 mm can be fabricated by using the AlMg35-TiH_2_ composite FA, as shown in Figure 4. Compared with the ball milling method, this one is more suitable for commercial production.

### 2.2. Gas Injection Foaming Method

In the early 1990s, Alcan company in Canada discovered that foams could be produced by injecting gas into liquid aluminum matrix composites, and the GIF method was developed [3]. The process is shown in Figure 5a. Bubbles are blown into aluminum alloy melt containing ceramic particles (usually SiC or Al_2_O_3_). Bubbles rise, accumulate, and solidify, and aluminum foams are fabricated [56]. Porosities of foams fabricated by this method are generally 75~98%, the pore size is generally 3~25 mm [8,57], and under certain conditions, foams with $d_{r}$ of 1 mm can be obtained [58,59]. The GIF method is suitable for the continuous production of slabs. The advantages of this method are its simple process, low cost, and continuous production [3,56]. Disadvantages are that it is difficult to disperse ceramic particles in the melt, blowing efficiency is low, pore size is large, and MEPs of products are poor.

In order to design a reasonable foaming process, research concerning the stability of foam has been carried out. The main factors affecting foam stability are ceramic particles and oxide film on the cell wall surface [61,62,63,64,65]. Directly blowing gas into pure aluminum or aluminum alloy melt cannot foam, there must be certain amounts of ceramic particles in the melt to foam [66]. Leitlmeier et al. [56] found that when the melt contains certain amounts of ceramic particles, the bubbles need to rise for a certain distance to obtain a stable foam structure. The minimum bubble traveling distance is inversely related to the volume fraction of ceramic particles in the melt. Liu [67] analyzed the above phenomenon and found that certain amounts of ceramic particles must be adsorbed on the bubble surface during the bubble-rising process. To obtain stable foam, the particle coverage ratio on the bubble surface must be greater than 14%. The influence of particle type, size, and content on foam stability was systematically studied in the reference [68,69], and requirements for the content and size of ceramic particles were concluded in the reference [8,69]. When the volume fraction of ceramic particles is between 5~20% and the size is 1~20 μm, stable foams can be obtained. However, no stable foam can be obtained when blowing inert gas into a melt containing ceramic particles [63,64]. Zhou [64] found that stable foams can be fabricated only when the oxygen volume fraction of gas is larger than 1.6%. An oxide film about 15 nm thick covers the entire bubble surface. The oxide film avoids particles piercing the surface of liquid films and limits the flow direction of drainage, and, when foams are deformed at the accumulation stage, the oxide film can protect the foams [65].

For the fabrication process, the design of nozzles, blowing mode, and base metal were studied to optimize pore structure. In terms of the design of nozzles, Liu [70] and Yuan [71] found that gas chamber volume, orifice diameter, and gas flow rate are important factors affecting bubble size. Bubble size can be effectively reduced by reducing gas chamber volume, orifice diameter, or gas flow rate (A gas chamber is defined as the space between the nozzle tip and the point where the pressure drop is large [72]). When the gas chamber volume is the same, gas injection devices with more orifices can blow smaller bubbles. Yuan [60,73] studied the bubble formation process during the static GIF process. The influence of contact angle is elucidated. When the contact angle becomes larger, bubbles detach later, and bubble size becomes larger. The existence of a wedge angle on the orifice can reduce the equivalent contact angle and reduce bubble size. For the blowing modes, the vibration and movement of nozzles can effectively reduce bubble size. In the work of Babcsán [58], the use of ultrasonic vibration on the nozzle was shown to enable precise control of bubble size. Wang [74] used a high-speed horizontal oscillation system and reduced the pore size of foams from 10 mm to less than 4 mm. Based on this oscillation system, a bubble formation and detachment model [75] was established. According to the model, bubble size is inversely proportional to the oscillation frequency and amplitude, and is proportional to the gas flow rate. Noack et al. [59] found that rotation of the gas injector can effectively reduce the pore size of foams, and foams with $d_{r}$ below 1 mm can be obtained in this way. In terms of the base metal, aluminum foams fabricated by hypoeutectic Al-Si alloy have a better pore structure and foam stability, and the average pore size is smaller [76].

### 2.3. Powder Metallurgy Foaming Method

In the 1990s, the Fraunhofer Institute in Germany made a breakthrough in the fabrication process of the PMF method [3]. The process is to compact the mixed powder of metal and FA to obtain a condensed precursor, then heat the precursor above its melting temperature, the FA in the precursor releases gas to form a foam structure, and aluminum foam is obtained after cooling [77], as shown in Figure 6a. The PMF method can realize near-net shape forming, and shaped aluminum foam parts (SAFPs) can be fabricated without machining [77,78]. Additionally, the PMF method is suitable for the fabrication of aluminum foam sandwich (AFS) panels, as shown in Figure 6b. Because the metallurgical bonding between foams and metal panels can be achieved [79]. Banhart et al. [32,77,79] reported the commercial production of AFS panels. The disadvantage of this method is that the cost of metal powder is high, and when fabricating products with nonuniform thickness or large size, the precursor is difficult to heat uniformly, which may cause a nonuniform pore structure.

Densification of precursors, alloy compositions, selection of FAs, and foaming process are important factors during the fabrication process. The precursors have to be condensed before foaming to prevent gas loss during foaming [4]. Various techniques, extrusion, powder rolling, uniaxial, and isostatic pressing have been used for powder consolidation [77,82,83]. For alloy composition, early works concentrated on pure Al, and it was soon discovered that alloys have advantages over pure Al [3]. Al-Si [84], Al-Si-Cu [85], Al-Mg-Si [86], Al-Sn [87], Al-Si-Mg-Cu-Sn [88], and other systems were developed. In these systems, AlSi_6_Cu_4_, AlSi_6_Cu_6_, and AlSi_8_Mg_4_ are more commonly used [79]. FAs mainly include TiH_2_ [77], CaCO_3_ [89], ZrH_2_ [90], Mg-Al alloy powders [91], LiAlH_4_ [92], etc. TiH_2_ has the best foaming result. Pre-oxidation treatment can retard the decomposition of TiH_2_ and has a beneficial effect on pore morphology [49]. The cell structure of foams can be improved by pretreating TiH_2_ with a layer of Sn powder [93]. With regard to the foaming process, heating and cooling rates have an important effect on the foaming process. Insufficient heating rates during foaming, <1 K/s, may cause cracks in the precursor and gas loss [94], while at higher heating rates, the precursor might not be heated uniformly, resulting in inhomogeneous foaming [95]. During the cooling process, increasing the cooling rate (>2.5 K/s) can mitigate drainage and bubble coalescence and reduce the number of defects in the cell wall. However, when the cooling rate is too high, it can introduce thermal shock to the foam, causing cell rupture [96]. In recent years, the development of synchrotron technology enables real-time observation during the fabrication of aluminum foams. Nucleation, growth of bubbles [97,98], and evolution of pore structure [99,100,101,102,103] have been investigated, which can guide optimization of the fabrication process.

## 3. Processing

The reason that metallic materials are commonly used is not only their properties, but also numerous processing methods [104]. Although great progress has been made in the fabrication techniques, the wide use of aluminum foams still faces challenges. The porosity of aluminum foams is usually over 50%, and porous structures seriously limit their processing behavior [104,105]. The pore structure of foams collapses when it is compressed, so it is difficult to process through plastic deformation. Foam melt with high viscosity is difficult to flow, and not suitable for conventional casting processes. Due to the low proportion of metal and oxidation on the interface, the weldability of aluminum foams is poor [106]. SAFPs are usually processed by machining, which leads to high cost and low efficiency.

During the development process of aluminum foams, many techniques have been tried for processing. Some of the most important processing techniques, PMF foaming process, two-step foaming (TSF) process, cast foaming (CF) process, GIF process, mold pressing (MP) process, and integral foaming (IF) process are summarized here.

### 3.1. Powder Metallurgy Foaming Process

SAFPs can be produced by using the PMF process, as shown in Figure 7. When the precursor in the mold is heated, the FAs in the precursor decompose and the precursor expands and fills the mold [80], as shown in Figure 6a. However, this process has some limitations. When producing products with large size or nonuniform thickness, it is difficult to heat the precursor uniformly, which may cause a nonuniform pore structure. Additionally, the mold-filling ability of solid precursor is not good enough, sometimes the mold cannot be fully filled [107]. In addition, the price of metal powder is high and leads to high product costs.

### 3.2. Two-Step Foaming Process

The TSF process is to add FA TiH_2_ into the thickened melt, quickly disperse the FA, and cool the melt to obtain a solid precursor, then heat the precursor in a mold to obtain SAFPs [108,109], as shown in Figure 8. The TSF process has similar problems to the PMF process. It is difficult to heat the precursor uniformly, and the mold-filling ability of the precursor is not good enough. In addition, generally, there are many pores in the precursors, pores reduce the thermal conductivity of precursors and worsen the foaming result. According to the experimental result of Shang et al. [108], good foaming results can be obtained only when the precursor porosity is lower than 56%.

### 3.3. Cast Foaming Process

Yuan [104] developed the CF process on the basis of the MF method, the process is shown in Figure 9a. Near eutectic Al-Si-Mg alloy with a low melting point (557 °C) is used as the precursor alloy. Add pre-oxidized TiH_2_ FA into the thickened Al-Si-Mg melt at ~600 °C, and stir to disperse the TiH_2_ powders uniformly into the melt. Pre-oxidized TiH_2_ will not decompose rapidly at 600 °C, and the precursor melt can be cast into a mold. The precursor melt is heated by the 700 °C high-temperature mold, TiH_2_ releases gas, melt expands and SAFP is obtained. The mold-filling ability of liquid precursor is good, and complex shape SAFPs can be produced, as shown in Figure 9b. Near net-shaped parts with dense skin can be produced.

Metallic molds are not suitable for the CF process, because repeated heating and cooling can cause deformation or even damage to metallic molds. The application of the CF process through investment casting provides the opportunity for commercialized production of SAFPs [104].

But there are still some problems with this process. In order to ensure that the precursor melt can be cast into the mold, and fit the decomposition temperature of FA TiH_2_, only aluminum alloys with a low melting point can be used. The eutectic Al-Si-Mg alloy, which is not an appropriate alloy for foam fabrication, is needed in this process. The stability of Al-Si-Mg foams is worse than Al-Ca foams. Surface shrinkage and inhomogeneous pore structure defects may appear. Therefore, this process is only suitable for the production of small-size parts.

### 3.4. Gas Injection Foaming Process

Cingi et al. [110] combined the investment casting process with the GIF process to produce SAFPs, as shown in Figure 10a–c. Aluminum foam parts with dense skin can be obtained in this way. The limitation of this process is that only simple shape parts can be fabricated, and pore structure control is difficult.

Babcsán et al. [58,111] reported the production of ALUHAB aluminum foam, sub-millimeter pore size foams. According to the literature [58], ALUHAB foams have an extremely stable pore structure, which can be re-melted or cast into complex shapes without damage to the foam structure, as shown in Figure 10d. However, nano-scale particles have to be dispersed in the melt before foaming, which is difficult to achieve in commercial production.

### 3.5. Mold Pressing Process

Filice et al. [112] and Banhart et al. [32] found that aluminum foams or their composite structures can be mold pressed or forged to produce SAFPs, as shown in Figure 11a,b.

Zhang et al. [113] found that AFS panels can keep their pore structure during MP deformation, as shown in Figure 11c. The risk of core fracture can be reduced by adopting multi-step forming and increasing the core relative density.

Liu [114] found that aluminum foams can deform during the solid–liquid–gas coexisting state and shape formation can be achieved through MP, as shown in Figure 11d–g. Characteristic parameters were almost unchanged during formation.

**Figure 11 materials-17-00560-f011:**
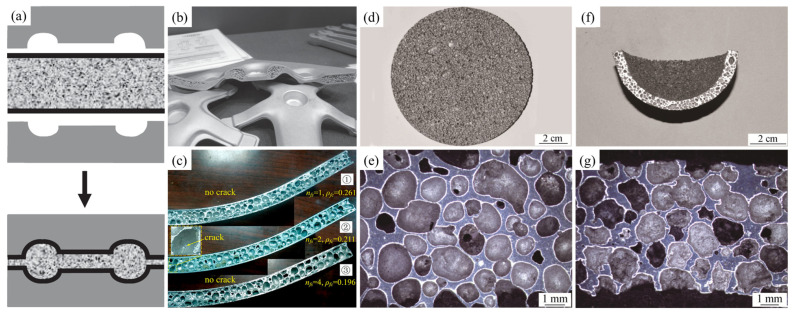
(**a**) MP of AFS panel [32] (reproduced with permission from John Wiley and Sons); (**b**) mold pressed AFS parts [32] (reproduced with permission from John Wiley and Sons); (**c**) deformed AFS panels [113] (reproduced with permission from Elsevier); (**d**–**g**) macroscopic morphologies and pore structure before and after MP [114] (reproduced with permission from Springer).

However, this process still has some limitations. MP process is more suitable for the deformation of aluminum foam panels. It is still difficult to realize the deformation of aluminum foams with big sizes or complex shapes. When the deformation is large, the pore structure of foams might be destroyed or even condensed. In addition, when aluminum foams are heated at high temperatures, oxidization will happen on the surface and defects might appear.

### 3.6. Integral Foaming Process

The IF of aluminum is similar to methods of producing polymer integral foams [105,115,116,117], as shown in Figure 12. MgH_2_ FA with a relatively low decomposition temperature is put in the runner and gating system. Liquid metal is injected with high velocities into permanent steel molds. During this stage, FA is dispersed into the melt. The FA releases gas and the melt begins to foam, and, at the same time, solidification of the melt happens at the surface of the mold. Near net-shaped foam parts with dense skin can be obtained in this way. However, control of sample pore structure is difficult in this process.

### 3.7. Summary

The above-mentioned processes produced some prototypes, but there is still a big gap between real production. Processing techniques of aluminum foams cannot be connected with fabrication methods. Many processing techniques, such as the TSF process and CF process, are not practical for fabrication, limitations of these processing methods are low efficiency, poor shaping ability, or difficulty of pore structure control. The MF method, which is suitable for the commercial production of large-size aluminum foams, is not suitable for producing SAFPs.

Processing of aluminum foams is still difficult. If a practical processing technique with good shaping ability, uniform pore structure, and high efficiency can be developed, it will be of great importance to the further development of aluminum foams.

## 4. Properties and Applications

In this chapter, the properties and applications of aluminum foams are introduced, and the MEPs are emphatically elucidated.

### 4.1. Mechanical Properties

#### 4.1.1. Basic Concepts

The unique deformation behavior of aluminum foams during compression determines that it is suitable for energy absorption and crash protection [25]. Deformation during compression can be divided into three stages [118], as shown in Figure 13. In the first stage, elastic deformation occurs. In the second stage, the plateau stress stage, with the increase in strain ε, the stress σ almost remains constant. In the third stage, densification stage, σ increases sharply with the increase in ε. When aluminum foams are used for protection, energy can be absorbed while keeping the stress of protected objects below a relatively low level [2].

The main parameters of aluminum foams, porosity P, relative density $\rho_{s}$, average pore size $d_{r}$, densification strain $\varepsilon_{d}$, plateau stress $\sigma_{p}$, and energy absorption at densification strain $W_{d}$ can be calculated according to the equations below:(1)$$P=1-\frac{\rho^{*}}{\rho_{Al}}$$
where $\rho_{Al}$ is the density of bulk Al, and $\rho^{*}$ is the density of aluminum foam. Relative density:(2)$$\rho_{s}=\frac{\rho^{*}}{\rho_{Al}}=1-P$$

The average pore diameter can be calculated as [119]:(3)$$d_{r}=\frac{d_{s}}{0.785}=\frac{\sum_{i=1}^{N}d_{i}}{0.785N}$$
where $d_{r}$ is the average pore diameter of aluminum foam, $d_{s}$ is the average diameter of pores on the 2D sample image,$\;d_{i}$ is the equivalent diameter of each pore, and N is the number of pores.

Energy absorption during deformation is
(4)$$W=\int_{0}^{\varepsilon }\sigma d\varepsilon$$

Energy absorption at densification strain $\varepsilon_{d}$ is
(5)$$W_{d}=\int_{0}^{\varepsilon_{d}}\sigma d\varepsilon$$

The unit of W and $W_{d}$ is MJ/m^3^. According to the equations above, it is obvious that aluminum foams with higher values of $\sigma_{p}$ or $\varepsilon_{d}$ have higher $W_{d}$.

According to the reference [120], the value of $\varepsilon_{d}$ depends on the energy absorption efficiency η.
(6)$$\eta =\frac{W}{\sigma }=\frac{\int_{0}^{\varepsilon }\sigma d\varepsilon }{\sigma }$$

Before the densification stage, the increase rate of W is greater than σ, and η increases with the increase in ε. In the densification stage, strain hardening becomes obvious. The increase rate of σ exceeds W, and η will decrease, as shown in Figure 13a. The ε corresponding to the turning point of the η curve is $\varepsilon_{d}$.
(7)$$\frac{d\eta }{d\varepsilon }{|}_{\varepsilon =\varepsilon_{d}}=0$$

Sometimes there is a slope during the plateau stress stage and the equivalent plateau stress $\sigma_{p}$ is often used.
(8)$$\sigma_{p}=\frac{\int_{\varepsilon_{0}}^{\varepsilon_{d}}\sigma d\varepsilon }{\varepsilon_{d}-\varepsilon_{0}}$$

In real occasions, the stress reached during compression cannot be too high. Otherwise, damages or injuries might be caused [26]. So, sometimes using the σ-W diagram instead of ε-W diagram is more suitable.

Quasi-static compressive test (QSCT) results [26] of aluminum foams are shown in Figure 14. It can be seen from Figure 14a,b that σ and W increase monotonically with the increase in $\rho_{s}$ at the same ε level. However, aluminum foams with higher $\rho_{s}$ do not always possess higher value of W at the same σ level. This can be seen in Figure 14c,d.

Aluminum foams are often used for weight reduction and energy absorption. Therefore, in practical applications, it may be a concern how much energy aluminum foam with a given weight can absorb. Zhou [27] proposed the concept of energy absorption per unit mass $W^{*}$ (unit: J/g), Equation (9), and energy absorption efficiency per unit mass $\eta^{*}$, Equation (10).
(9)$$W^{*}=\frac{W}{\rho^{*}}=\frac{\int_{0}^{\varepsilon }\sigma d\varepsilon }{\rho^{*}}$$
(10)$$\eta^{*}=\frac{\eta }{\rho^{*}}=\frac{\int_{0}^{\varepsilon }\sigma d\varepsilon }{\rho^{*}\sigma }$$

The QSCT results [27] are shown in Figure 15. The $W^{*}$ of the aluminum foam with the lowest $\rho_{s}$ is not always the highest, as shown in Figure 15d. And it can be seen from Figure 15e that each $\eta^{*}$ curve has a peak value. The peak value decreases with the increase in the $\rho_{s}$. Comparing Figure 15a,d,e, it can be found that when the porosity is lower than 55%, the peak value of $\eta^{*}$ is only about 0.2. If the porosity is further reduced, $W^{*}$ is almost the same, but the stress σ increases sharply. This indicates that, although samples with higher $\rho_{s}$ have higher $\sigma_{p}$ and $W_{d}$, they may not be suitable for energy absorption or crash protection.

When used for energy absorption or crash protection, it is not always the case that aluminum foams with higher $\rho_{s}$ or $\sigma_{p}$ have better energy absorption ability. For samples with different $\rho_{s}$, there exists an optimal σ value, under which each sample can absorb the maximum energy. Therefore, it is necessary to clarify the optimized use conditions of aluminum foams and find out the most suitable application.

Different from QSCT conditions, gas trapped in the cell wall increases the $\sigma_{p}$ and W of closed-cell aluminum foams at higher strain rates $\dot{\varepsilon }$ [121,122,123,124,125]. According to the result of Paul et al. [126], the $W_{d}$ of ALPORAS foams increased by more than 50% when the $\dot{\varepsilon }$ increased from 3.3 × 10^−5^ s^−1^ to 1.6 × 10^−1^ s^−1^. This means that aluminum foams are suitable for crash protection.

#### 4.1.2. Influencing Factors

The MEPs of aluminum foams are mainly influenced by pore structure, cell wall defects, and microstructure [127,128,129,130,131,132,133].

Pore Structure

The pore structure of aluminum foams is of great importance to their MEPs, and the specific factors are relative density $\rho_{s}$, pore morphology, and homogeneity of pores. Yang et al. [134] found that with the increase in $\rho_{s}$ from 0.08 to 0.36, $\sigma_{p}$ increased by more than 10 times, and $W_{d}$ increased by more than 7 times. This indicates that the MEPs of aluminum foams can be controlled or adjusted within a large range by adjusting the $\rho_{s}$ to meet the requirements of different applications. Sugimura et al. [135] studied the influence of cell morphology on its MEPs and found that samples with less curved and serrated cell walls exhibit higher stiffness. Zuo et al. [54] found that the $\sigma_{p}$ of Al_9_Si foams with the same porosity of 75% doubled when $d_{r}$ decreased from 3 mm to 1 mm [54]. With the decrease in $d_{r}$, there are more pores per unit volume, and structural defects are dispersed.

Based on a lot of research work, control of pore structure has been realized (introduced in Section 2.1, Section 2.2 and Section 2.3), and the MEPs of aluminum foams have been greatly improved, as shown in Figure 4. For small pore foams with $d_{r}$ of 1~2 mm and porosity of around 60~70%, $\sigma_{p}$ increased to about 15~30 MPa and $W_{d}$ increased to about 10~15 J/cm^3^.

Defects in the Cell Wall

During the cooling and solidification stage of foam fabrication, defects such as cell wall broken and shrinkage will inevitably occur [136,137,138]. These defects are harmful to the MEPs. In real conditions, isolated pore structures cannot be obtained. According to the reference [137], for MF aluminum foam with a porosity of 77%, more than 90% of the pores have cell wall broken defects, and for GIF aluminum foam with a porosity of 86%, more than 30% of the pores have cell wall broken defects. In addition, there are many micropores in the cell wall [136].

Hu [139] found that after water spray chilling, the proportion of broken cell walls is significantly reduced, as shown in Figure 16. The number of micropores is also reduced, as shown in Figure 17.

Microstructure and Properties of the Cell Wall

The cell wall is the stressed portion and cell wall properties directly determine the properties of aluminum foams [130,140]. Figure 18 summarizes the relationship between $\rho_{s}$ and $\sigma_{p}$ of MF foams with different compositions. It can be seen that the $\sigma_{p}$ of samples with similar $\rho_{s}$ has a big difference that can be more than double. This indicates that the microstructure of aluminum foams has a big influence on their properties.

When the fabrication process is stable and the pore structure is controllable, the microstructure and properties of the cell wall become the most important factor. In addition, improving the MEPs of aluminum foams by reducing $d_{r}$ may lead to an increase in $\rho_{s}$ and the decrease in $\varepsilon_{d}$, which is bad for energy absorption and protection, while modification of cell wall microstructures does not have such problems.

**Figure 18 materials-17-00560-f018:**
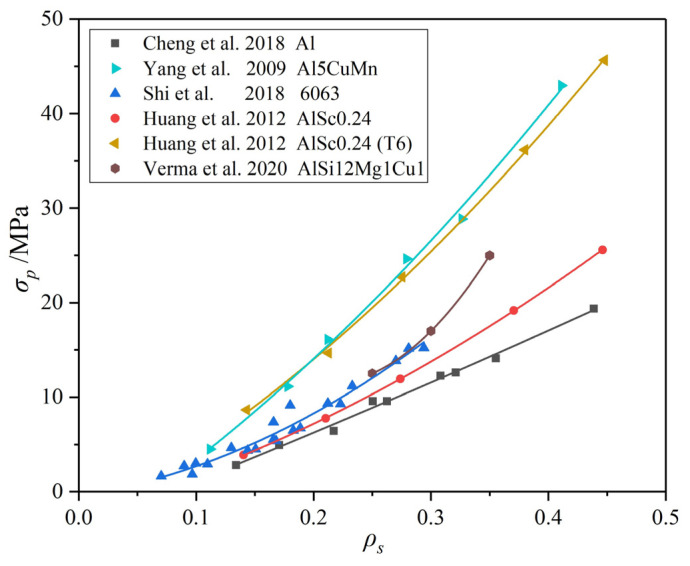
Relationship between $\rho_{s}$ and $\sigma_{p}$ of MF aluminum foams with different compositions [26,134,141,142,143].

The microstructure of aluminum foams is quite different from conventional bulk Al alloys. Aluminum melt has to be treated before foaming [81]. For example, the MF method requires the thickening process before foaming [36], and in the process of the GIF method, ceramic particles have to be dispersed in the melt before foaming [66]. Hu [144] characterized the microstructure of MF foams by the improved anodization method, as shown in Figure 19c–f.

The solidification of aluminum foams and conventional bulk alloys are very different. Hu [144] proposed the solute diffusion model of growing α-Al grains in foamed melt restricted by surrounding bubbles, as shown in Figure 19a,b. In foamed melt, the solute diffusion of grain growth is limited by the bubbles, resulting in solute enrichment at the solid–liquid interface, thus the grain size of aluminum foams is smaller than bulk aluminum alloys under the same conditions. Grains in aluminum foams will replicate the contours of cell walls or plateau borders, presenting irregular shapes, as shown in Figure 19c,e. The grain morphology of aluminum foams can be modified by increasing the cooling rate or inoculation treatment, as shown in Figure 19d,f. After modification of cell wall microstructures by increased cooling rate and inoculation, the strength and toughness of aluminum foams can be improved simultaneously [139].

After decades of research, aluminum foams with different alloy compositions have been fabricated. For example, MF pure Al [36], Al-Cu [145], Al-Mn [146], Al-Zn [147], Al-Sc [142] systems, GIF Al-Si-Mg [57], Al-Si-Cu [56] systems, PMF Al-Si [84], Al-Si-Cu [85], Al-Mg-Si [86], Al-Sn [87], and Al-Si-Mg-Cu-Sn [88] systems. Heat treatment [145,148,149], matrix composite [150,151,152], and other methods have been used for strengthening.

Although much research on the microstructure and properties of aluminum foams have been reported, there is still room for improvement. During the fabrication process of aluminum foams, priorities are usually given to the foamability of the melt and pore structure control, while the microstructure is relatively less concerned. Some research works directly use a certain type of aluminum alloy to foam. However, most aluminum alloys are not developed for the fabrication of foams. After foaming, the microstructure of foams is significantly different from the original bulk alloy. In addition, heat treatment is necessary for many high-strength alloys, but heat treatment may destroy the pore structure of aluminum foams. These problems limit the MEPs of aluminum foams. The MEPs of aluminum foams can be further improved if the cell wall microstructure can be modified while keeping a uniform pore structure.

#### 4.1.3. Problems

At present, the MEPs of aluminum foams are not good enough, and there are still some problems. First, there is a gap between the predicted and the tested results. Second, the MEPs of aluminum foams are not stable and difficult to control within a reasonable range.

Various methods, such as theoretical model prediction [153], finite element simulation [154], and machine learning [155] have been used to predict the mechanical property parameters of aluminum foams. Here, the Gibson–Ashby model [153], one of the most commonly used theoretical models, is used to predict the plateau stress of aluminum foams under idealistic conditions. According to the reference [156], the plateau stress of tetrakaidekahedral simplified closed-cell foams is
(11)$$\frac{\sigma_{p}^{*}}{\sigma_{s}}=0.33{\left(\frac{\rho^{*}}{\rho_{Al}}\right)}^{2}+0.44\left(\frac{\rho^{*}}{\rho_{Al}}\right)$$
where $\sigma_{p}^{*}$ is the predicted plateau stress of aluminum foams, $\sigma_{s}$ is the yield strength of bulk aluminum alloys, $\rho^{*}/\rho_{Al}$ is the relative density of aluminum foam. For foams with a relative density below 0.2, the influence of ${\left(\rho^{*}/\rho_{Al}\right)}^{2}$ can be neglected, and the equation can be simplified as
(12)$$\frac{\sigma_{p}^{*}}{\sigma_{s}}\approx 0.44\left(\frac{\rho^{*}}{\rho_{Al}}\right)$$

The yield strength of cast aluminum alloys is about 130 MPa [157] and 250 MPa for aluminum alloys with higher Si content and hardness [135], so the $\sigma_{p}^{*}$ can be calculated:(13)$$\sigma_{p}^{*}\approx 57\sim 110\left(\frac{\rho^{*}}{\rho_{Al}}\right)\;\mathrm{MPa}$$

Figure 20 compares predicted results with tested results in the recent literature. It can be seen that the tested result is lower than the predicted result, which means the MEPs of aluminum foams still have room for improvement.

**Figure 20 materials-17-00560-f020:**
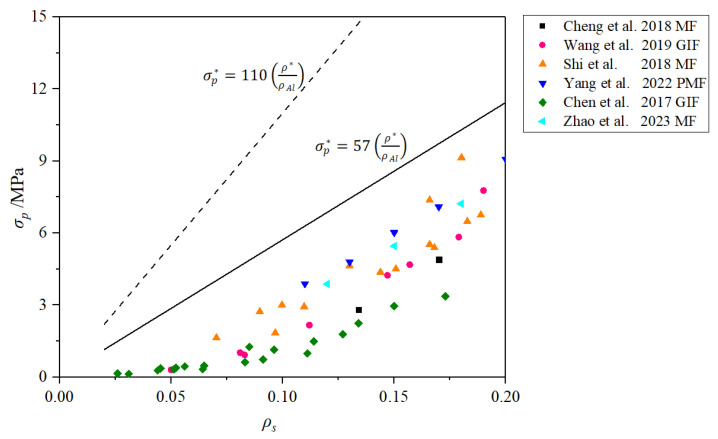
Comparison of tested and predicted $\sigma_{p}$ of aluminum foams [26,57,141,158,159,160].

In addition to the large difference between predicted and tested results, the MEPs of aluminum foams are not stable. According to Figure 20, for samples with similar $\rho_{s}$, the gap of $\sigma_{p}$ can be more than 100%. This is because foams fabricated by different methods have different pore structures and microstructures. For foams with the same porosity, mechanical property parameters, such as $\sigma_{p}$, $W_{d}$, cannot be clarified within a certain range, and there is no standard or grade to regulate the MEPs of aluminum foams at present. This is not conducive to their applications.

#### 4.1.4. Applications

Figure 21 shows some cases of aluminum foams used for protection. In Figure 21a,b, AFS panels are used on the high-speed train. In Figure 21c,d, aluminum foams and their composite structure are used in automobiles. Figure 21e–g show the application of aluminum foams in different protective occasions.

In practical applications, in addition to the direct use of aluminum foams, aluminum foam composite structures are often used [161]. Composite structures with metal sheets or tubes can be fabricated by PMF, adhesion, and other techniques. Compared with bare foams, composite structures have higher strength and MEPs [162]. However, this is not the main topic of this paper. Detailed descriptions of the fabrication and properties of aluminum foam composite structures can be found in recent reviews [161,162].

**Figure 21 materials-17-00560-f021:**
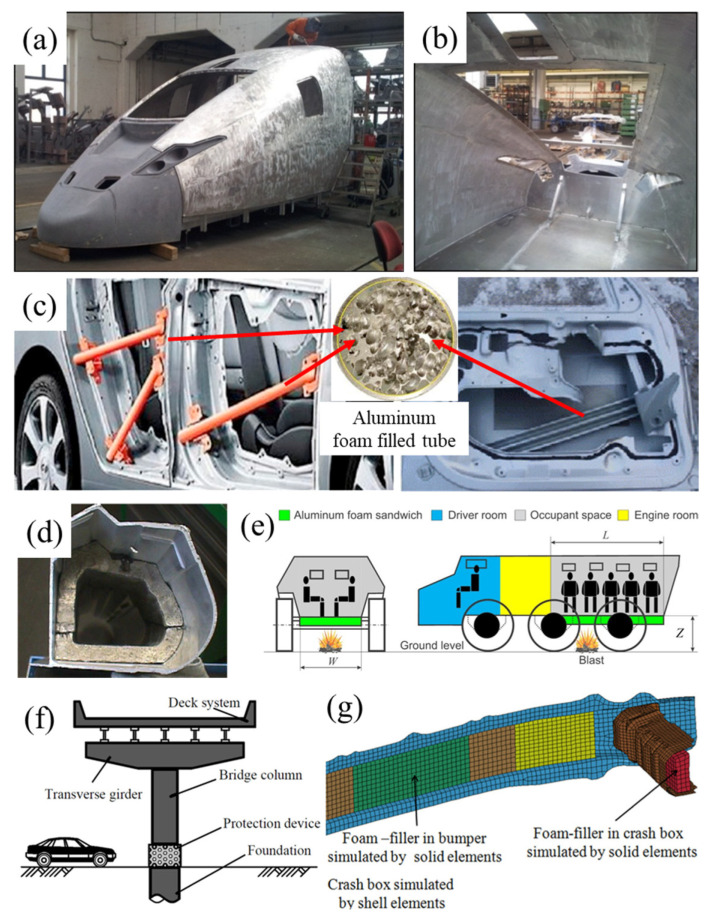
(**a**,**b**) German high-velocity train made of welded AFS [11] (reproduced with permission from MDPI); (**c**,**d**) automobile use [163,164] (reproduced/adapted with permission from Elsevier); (**e**–**g**) application in different protective occasions [165,166,167] (reproduced with permission from Sage and Elsevier).

### 4.2. Physical Properties

#### 4.2.1. Sound Absorption and Applications

Aluminum foams have good sound absorption properties. The surface of foams is not flat, hence, when a sound wave is transmitted to the surface, the wave diffusely reflects, and its intensity is weakened. After drilling holes, the connected cells of aluminum foams form a Helmholtz resonant sound absorption structure [168], which creates strong friction between the intensely vibrating air and the connected cells, and results in more sound energy consumption [37]. It can effectively absorb sound waves with frequencies of 250~5000 Hz [37,169]. Sound absorption of closed-cell aluminum foams can be improved by drilling holes [170], setting a cavity at the back of aluminum foams [22], or a certain degree of compression [171].

Compared with traditional sound-absorbing materials, like rock wool board and polymers, aluminum foams are environment friendly, non-toxic, and have the advantages of strong weather resistance and a beautiful appearance [37,38,168]. Aluminum foams have been used in architectural decoration and road traffic occasions for sound absorption, as shown in Figure 22.

#### 4.2.2. Electromagnetic Shielding and Applications

When an electromagnetic wave is transmitted to the surface of aluminum foams, reflection loss, absorption loss, wave–current interaction, and eddy-current loss reduce its intensity [29,172,173]. Xu et al. [174] tested the electromagnetic shielding effectiveness of aluminum foam samples with porosities of 75~93%. The shielding effectiveness ranges from 25–75 dB within the frequency of 130 to 1800 MHz.

Compared with traditional metal wire mesh, aluminum foams have higher shielding effectiveness. Compared with metal sheets, they are easier to install and have the advantage of being lightweight [29]. A damping box made of AFS panels for electromagnetic waves is shown in Figure 23.

#### 4.2.3. Heat Insulation and Applications

There are a large number of pores inside aluminum foams. Because the volume fraction of metal is small, and the surface of foams is usually covered by oxides, their thermal conductivity is low [175,176]. According to the experimental data of Zhu et al. [30], the thermal conductivity of aluminum foams with a porosity of 80% is 3.2 W/m·K and 1.7 W/m·K for a porosity of 90%. The low thermal conductivity makes it suitable for heat insulation and related occasions. Figure 24 shows the use of aluminum foams for barbecue plates and cooking ovens.

## 5. Conclusions and Perspectives

The most important research directions of aluminum foams, fabrication, processing, and properties are summarized in this paper, and their applications are briefly introduced.

(1)The main fabrication methods of aluminum foams are the MF method, GIF method, and PMF method, and all these methods have been used in commercial production. The MF method is suitable for the fabrication of large-size blocks; the GIF method is suitable for the continuous production of foam slabs; and the PMF method is suitable for the fabrication of shaped parts or composite structures. Extensive research has led to the precise control of aluminum foam pore structure, enabling the production of foams with small pore sizes, uniform structures, and excellent properties.(2)The processing techniques of aluminum foams are introduced. Although various processes have been reported, many of them have failed to achieve commercial production due to poor pore structure, high costs, low efficiency, or the difficulty of fabricating large-size products. The difficulties in processing seriously limit the application of aluminum foams, making it of great significance to develop practical processing techniques.(3)Aluminum foams are suitable for energy absorption and crash protection. When used for protection, it is not always the case that foams with higher strength have better energy absorption ability. The MEPs of aluminum foams are mainly influenced by their pore structure, cell wall defects, and cell wall microstructure. Control of pore structure has been realized while the cell wall microstructure is relatively less concerned. Currently, the MEPs of aluminum foams are not good enough and still have room for improvement.(4)In addition to MEPs, aluminum foams have many other unique characteristics including sound insulation, electromagnetic shielding, and heat resistance. They have been utilized in various fields including architecture, transportation, etc.

As reviewed above, the fabrication methods of aluminum foams have become stable and pore structure control and commercial production have been achieved. However, the processing of aluminum foams is still very difficult, and their MEPs are not good enough. If practical processing techniques can be developed, or the MEPs of aluminum foams can be further improved, these will be of great importance to their applications.

## Figures and Tables

**Figure 2 materials-17-00560-f002:**
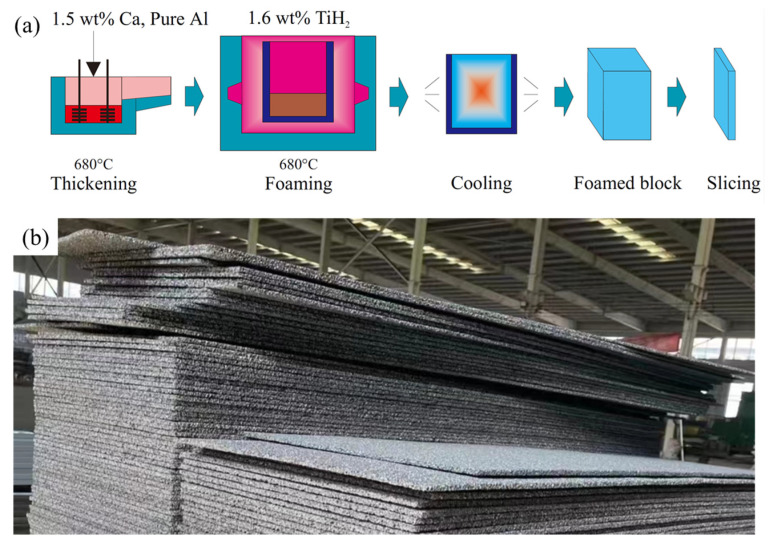
(**a**) The ALPORAS process [36] (reproduced with permission from John Wiley and Sons); (**b**) MF aluminum foam panels (reproduced with permission from Fumeite, Hebei, China).

**Figure 3 materials-17-00560-f003:**
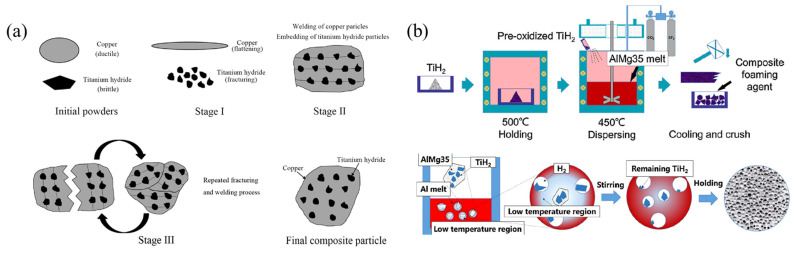
(**a**) Fabrication process of Cu-TiH_2_ composite FA [33] (reproduced with permission from Springer); (**b**) fabrication process of AlMg35-TiH_2_ composite FA and the foaming process [53] (reproduced with permission from Elsevier).

**Figure 4 materials-17-00560-f004:**
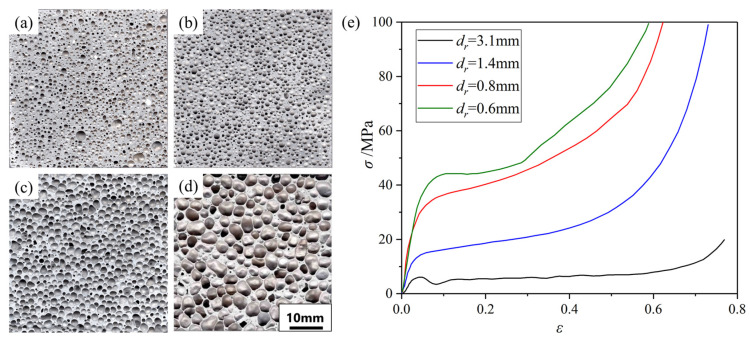
(**a**–**d**) MF aluminum foam samples with $d_{r}$ of 0.6, 0.8, 1.4, 3.1 mm fabricated by using the AlMg35-TiH_2_ FA [27] (reproduced with permission from Zhou Xu); (**e**) compressive properties of samples [27] (reproduced with permission from Zhou Xu).

**Figure 5 materials-17-00560-f005:**
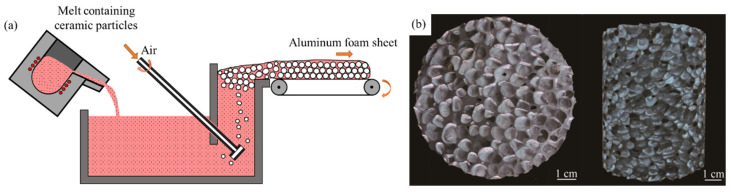
(**a**) Process of the GIF method; (**b**) samples [60] (reproduced with permission from Springer).

**Figure 6 materials-17-00560-f006:**
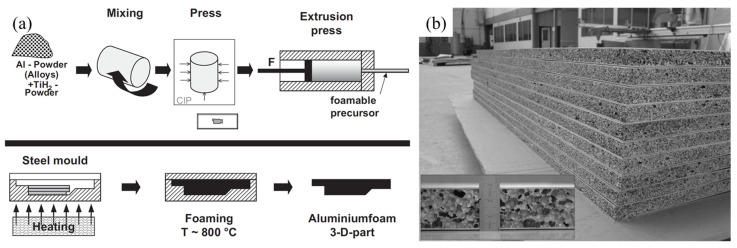
(**a**) Process of the PMF method [80] (reproduced with permission from John Wiley and Sons); (**b**) AFS panels [81] (reproduced with permission from John Wiley and Sons).

**Figure 7 materials-17-00560-f007:**
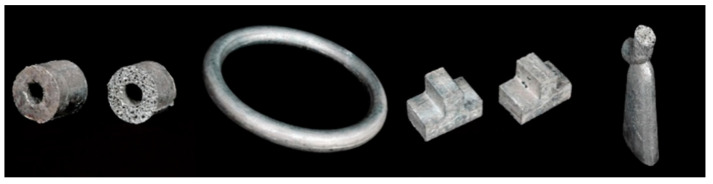
SAFPs produced by the PMF method [78] (reproduced with permission from MDPI).

**Figure 8 materials-17-00560-f008:**
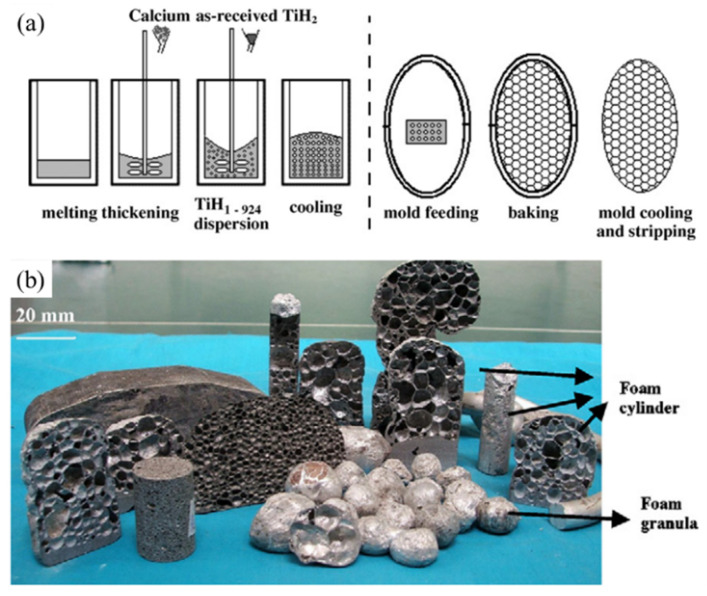
(**a**) The TSF process [108] (reproduced with permission from Elsevier); (**b**) parts foamed by using the TSF process [108] (reproduced with permission from Elsevier).

**Figure 9 materials-17-00560-f009:**
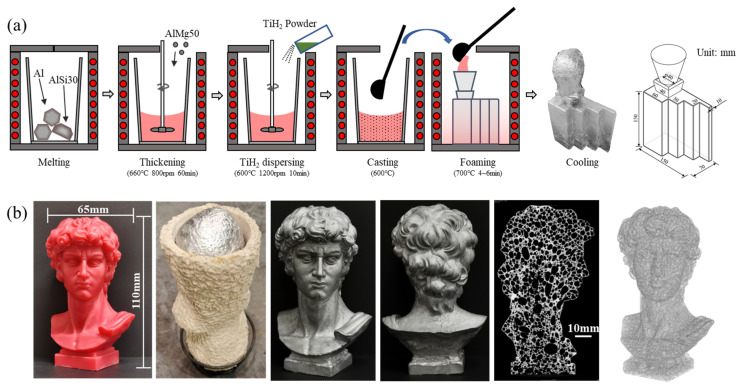
(**a**) The CF process [104] (reproduced/adapted with permission from Elsevier); (**b**) an aluminum foam portrait with complex shape produced by using CF process [104] (reproduced with permission from Elsevier).

**Figure 10 materials-17-00560-f010:**
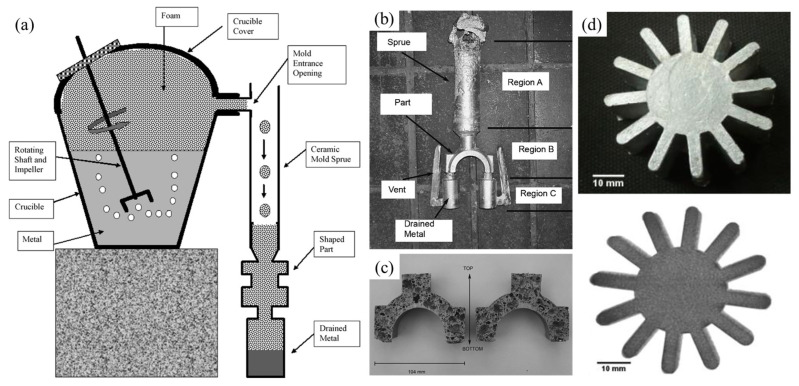
(**a**) The GIF process [110] (reproduced with permission from Elsevier); (**b**) a foamed part [110] (reproduced with permission from Elsevier); (**c**) pore structure of a foamed part [110] (reproduced with permission from Elsevier); (**d**) optical and X-ray photo of a cast ALUHAB part [111] (reproduced with permission from Elsevier.

**Figure 12 materials-17-00560-f012:**
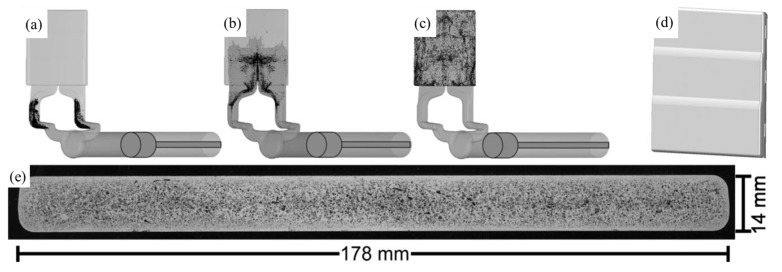
(**a**–**c**) Different steps of the IF process [116] (reproduced with permission from John Wiley and Sons); (**d**) geometry of the foamed part [116] (reproduced with permission from John Wiley and Sons); (**e**) pore structure of the foamed part [116] (reproduced with permission from John Wiley and Sons).

**Figure 13 materials-17-00560-f013:**
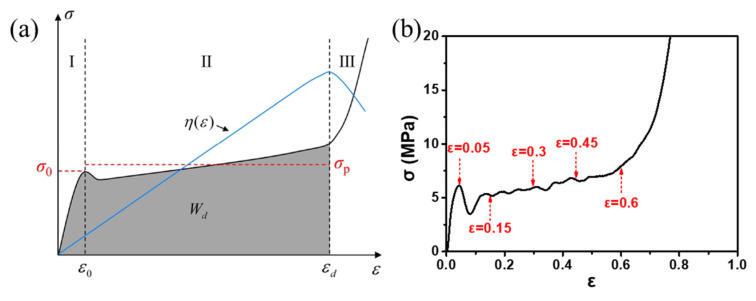
(**a**) Schematic diagram of the ε-σ and ε-η curve; (**b**) real ε-σ curve.

**Figure 14 materials-17-00560-f014:**
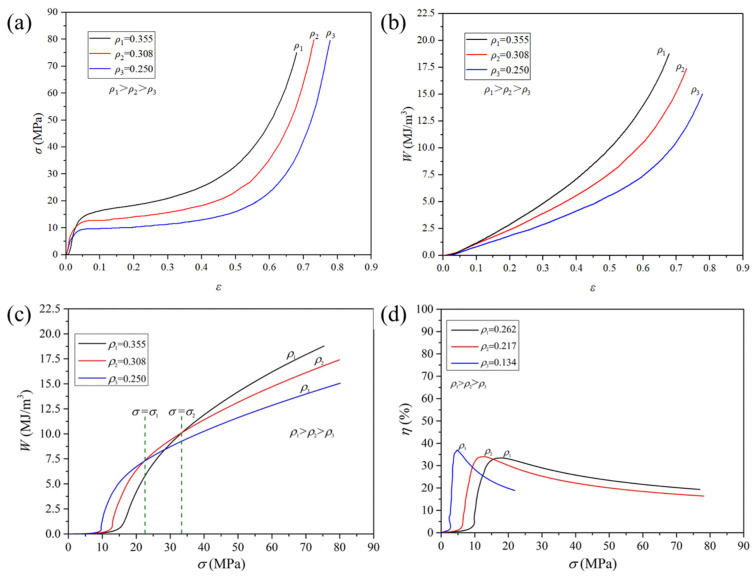
QSCT results of aluminum foams [26] (reproduced with permission from Springer) (**a**) ε-σ curve; (**b**) ε-W curve; (**c**) σ-W curve; (**d**) σ-η curve.

**Figure 15 materials-17-00560-f015:**
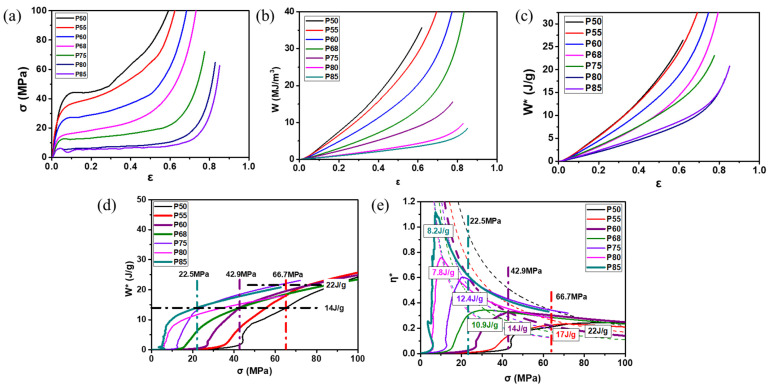
QSCT results of aluminum foams [27] (reproduced with permission from Zhou Xu) (**a**) ε-σ curve; (**b**) ε-W curve; (**c**) ε-$W^{*}$ curve; (**d**) σ-$W^{*}$ curve; (**e**) σ-$\eta^{*}$ curve; P represents porosity in figures, for example, P60 represents a sample with porosity of 60%.

**Figure 16 materials-17-00560-f016:**
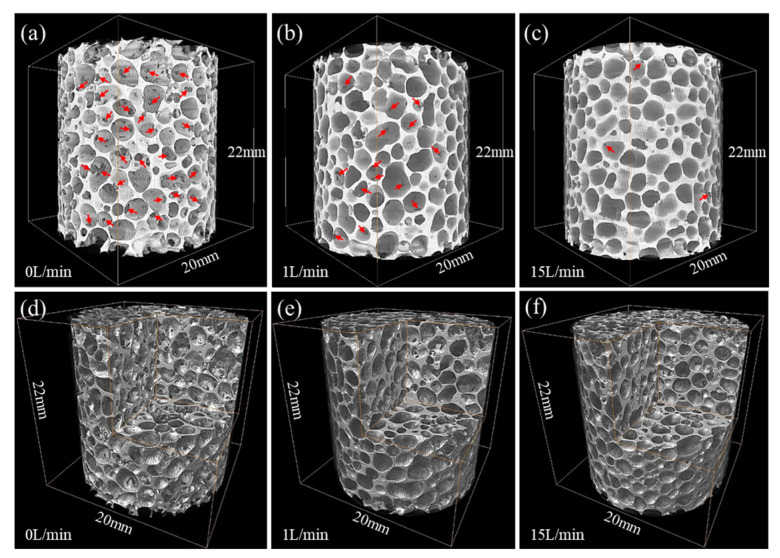
CT reconstruction image of MF aluminum foams under different volumes of water spray, red arrows point to the cell wall broken defects in the pictures [139] (reproduced with permission from Hu Lei) (**a**,**d**) 0 L/(m^2^·s); (**b**,**e**) 3.7 L/(m^2^·s); (**c**,**f**) 15.7 L/(m^2^·s).

**Figure 17 materials-17-00560-f017:**
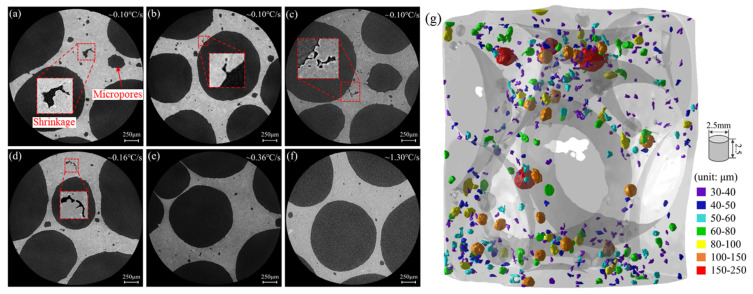
Micropores in the cell wall under different cooling rates [139] (reproduced with permission from Hu Lei) (**a**–**c**) 0.10 °C/s; (**d**) 0.16 °C/s; (**e**) 0.30 °C/s; (**f**) 1.30 °C/s; (**g**) CT reconstruction image of micropores, sample cooling rate 0.10 °C/s.

**Figure 19 materials-17-00560-f019:**
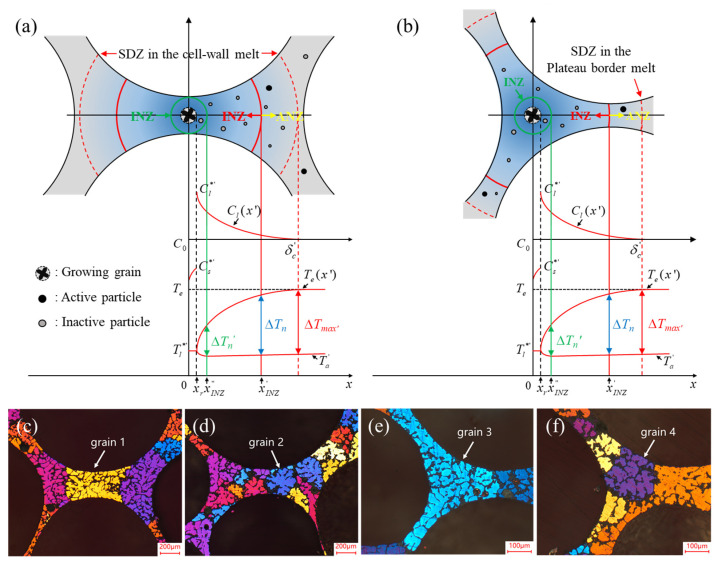
(**a**,**b**) Schematic diagram of the anisometric growth model of grains in the foamed melt [144] (reproduced with permission from Springer); (**c**,**d**) α-Al grains in the cell wall without and with inoculation [144] (reproduced with permission from Springer); (**e**,**f**) α-Al grains in the plateau border without and with inoculation [144] (reproduced with permission from Springer).

**Figure 22 materials-17-00560-f022:**
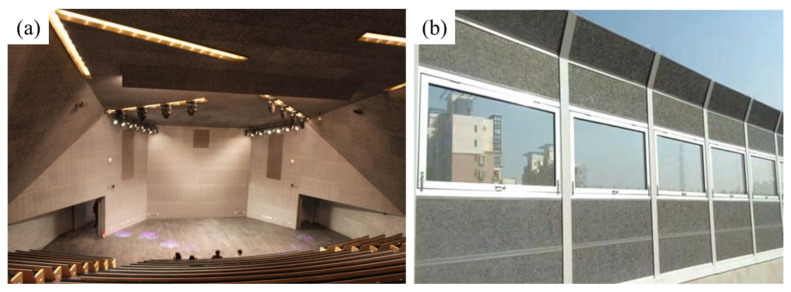
(**a**) The ceiling of an audience hall [11] (reproduced with permission from MDPI); (**b**) sound absorption barrier (reproduced with permission from Fumeite, Hebei, China).

**Figure 23 materials-17-00560-f023:**
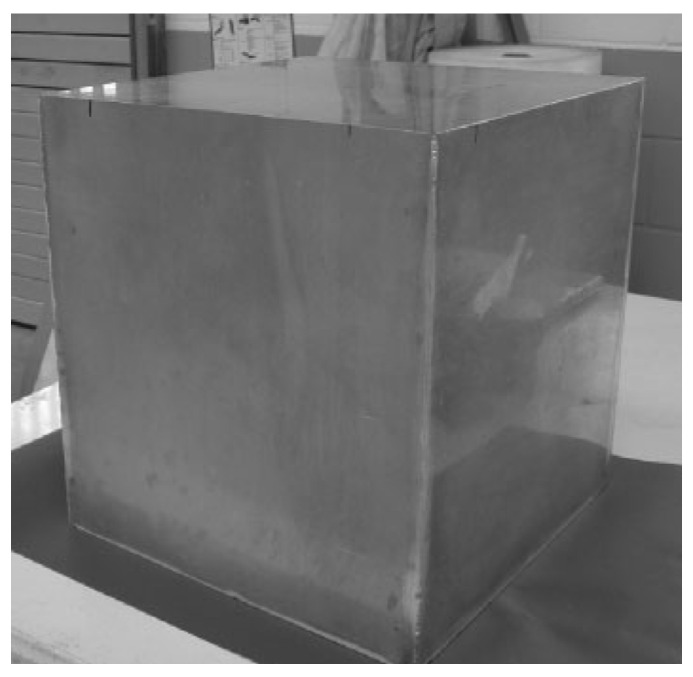
A damping box made of AFS panels for electromagnetic waves [79] (reproduced with permission from Jonh Wiley and Sons).

**Figure 24 materials-17-00560-f024:**
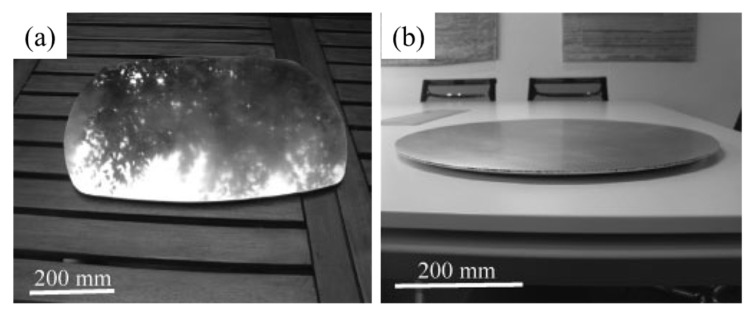
AFS panels for heat insulation [79]: (**a**) a barbecue plate; (**b**) a cooking plate for wood fire oven (reproduced with permission from Jonh Wiley and Sons).

**Table 1 materials-17-00560-t001:** Comparison of aluminum foams fabricated by the MF, GIF, PMF method.

Fabrication Methods	Pore Size	Porosity	Advantages	Disadvantages
MF	1~8 mm	50~90%	Fabrication of large-size blocks Low cost	Poor shaping ability
GIF	1~25 mm	75~98%	Simple process Continuous production Low cost	Difficult to disperse particles in the melt Blowing efficiency is low Pore size is large Poor mechanical properties
PMF	1~6 mm	50~90%	Near-net shape forming Fabrication of sandwich structures Metallurgical bonding between metal sheets	Pore structure control is difficult Large-size parts are difficult to make High cost

## Data Availability

Not applicable.

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
