# Peer review of "Fabrication, Processing, Properties, and Applications of Closed-Cell Aluminum Foams: A Review"

_materials, 2024, doi:10.3390/ma17030560_

Round 1

Reviewer 1 Report

Comments and Suggestions for Authors

line 11: "specific strength, big great energy absorption,"

line 62-63: suggested rewrite like "In decades of researches, many fabrication methods were proposed to get aluminum foam.

line 65-66:  "The research progress will be are reviewed in the following content."

line 69-70: "This ALPORAS process is shown in has been used until now, as shown in Figure 2 (a)."  With the sentence "as been used until us" do you mean that it will not be used in the future? are proposed some modifications?

line 83: "In Among these TAs, Ca has shows the best foam stabilization ability"

line 88: "In Among these FAs,"

line 88-96: rephrase the sentences.

line 94-95: "After the fabrication process of selecting Ca as the TA and preoxidized TiH2 as the FA is determined" - rephrase, for instance "Starting from Ca as the TA and TiH2 as FA, "

line 97: "Then realized batch fabrication of 500×1000×X mm slabs. " And ... which were the optimized parameters?

Line 107: "The wettability of TiH2 and in aluminum melt is poor"

line 115: "Foams with pore diameter 𝑑𝑟 of 1.6mm"  and line 120 and caption 4

line 122: "milling method, this method one"

Figure 4e fonts are too small

line 129: " liquid aluminum matrix composite melts"

--------------------------------------------------------

Comments on the Quality of English Language

The paper is not easy to read. It is sometime required to read twice the sentences, to try to "imagine" what the autors want to tell to the reader.

Reviewer 2 Report

Comments and Suggestions for Authors

I would like to commend the authors for their comprehensive review of closed-cell aluminum foams. However, I would like to suggest some improvements that could enhance the quality and impact of this paper.

1)       I recommend that the authors include an introduction section that provides a brief overview of the topic and its significance. This would help readers to better understand the context and importance of the research presented in the paper.

2)       I suggest that the authors consider including a table or figure summarizing the properties and characteristics of different types of aluminum foams. This would help readers to quickly compare and contrast the different types and better understand their unique features.

3)      I recommend that the authors include more references and citations to support the claims and conclusions presented in the paper. This would help readers to better evaluate the validity of the claims made in the review and to identify areas where further investigation is needed.

4)      Finally, I suggest that the authors add a discussion section that analyzes the results and conclusions presented in the paper and offers recommendations for future research. This would help readers to better understand the implications of the research and to identify areas where further investigation is needed.

Overall, I believe that these improvements would enhance the quality and impact of this paper and make it a valuable contribution to the field of closed-cell aluminum foams

Comments on the Quality of English Language

Minor editing of English language required

Reviewer 3 Report

Comments and Suggestions for Authors

A review has been done focusing on the most important aspects such as production, processing and properties.
The work is complete and I find it well structured with sufficient references.
However, I miss some studies where tools such as finite element modeling or neural networks are used to predict properties and behavior, which, as has been well reflected, is a difficult fact to predict and control. I will now present some papers in the bibliography that can help to complement this work.
Insights into building a digital twin of closed-cell aluminum foam during impact loading: Microstructural, experimental and finite element investigations: https://doi.org/10.1016/j.jmrt.2023.10.094
Prediction of compressive mechanical properties of three-dimensional mesoscopic aluminium foam based on deep learning method: https://doi.org/10.1016/j.mechmat.2023.104684
Finally, with regard to the conclusions, I feel that they are too brief for the review work that has been done. I would recommend that they be expanded
